# Extraction, Characterization and Antioxidant Activity in vitro of Proteins from *Semen Allii Fistulosi*

**DOI:** 10.3390/molecules23123235

**Published:** 2018-12-07

**Authors:** Min Zuo, Xiao-xiao Liu, Di Liu, Hang-yun Zhao, Lu-lu Xuan, Wen-xian Jiang, Wan-zhong Li

**Affiliations:** School of Pharmacy, Weifang Medical University, Weifang 261053, Shandong Province, China; ZuoMin1026@163.com (M.Z.); liuxx@wfmc.edu.cn (X.L.); ld0928928@163.com (D.L.); zhy132104@163.com (H.Z.); xuanll1995@163.com (L.X.); jiangwenxian123@163.com (W.J.)

**Keywords:** *Semen Allii Fistulosi*, protein, extraction, characterization, antioxidant activity

## Abstract

*Semen Allii Fistulosi* (PSAF) is the seed of *Allium fistulosum* L. of the Liliaceae family. The purpose of this study was to extract, characterize, and evaluate the antioxidant activity in vitro of proteins. Using single factor and orthogonal design, the optimum conditions of extraction were determined to be as follows: extraction time 150 min, pH 8.5, temperature 60 °C, and ratio (*v*/*w*, mL/g) of extraction solvent to raw material 35. The isoelectric point of the pH was determined to be about 4.4 and 10.2, by measuring the protein content of PSAF solutions at different pH values. The amino acid composition of PSAF was determined by high performance liquid chromatography (HPLC), and the results suggested that the species of amino acids contained in the PSAF was complete. Sodium dodecyl sulphate polyacrylamide gel electrophoresis (SDS–PAGE) analysis showed the molecular weight was mainly between 40 and 55 kDa, and Fourier-transform infrared spectroscopy (FTIR) characterized prevalent protein absorption peaks. PSAF exhibited potent scavenging activities against DPPH assays, via targeting of hydroxyl and superoxide radicals, while chelating Fe^2+^ activity and demonstrating weak reducing power. This work revealed that PSAF possessed potential antioxidant activity in vitro, suggesting potential for use of PSAF as a natural antioxidant.

## 1. Introduction

*Allium fistulosum* L. is a traditional Chinese vegetable and is one of the most commonly used vegetables in the daily life of Chinese families. The cultivated area of allium vegetables accounts for 10% of the total sown area of vegetables, and its yield accounts for 7% of the national agricultural output. *A. fistulosum* L. represents an important natural resource. *A. fistulosum* L. is an edible plant cultivated on a large scale, and according to the *Compendium of Materia Medica*, the roots, stems, flowers, and seeds of *A. fistulosum* L. could be used for medicinal purposes [1].

*A. fistulosum* L. has a variety of biological activities, such as antiseptic, anticancer, and antioxidative properties [2]. In particular, dry mature *A. fistulosum* L. seeds are often used for medicinal purposes, typically in the treatment of kidney deficiencies, vertigo, and other ailments [3]. *Semen Allii Fistulosi* extract is also known to prevent and treat myocardial ischemia [4]. Compounds imparting these effects are urgently needed [5], underlining the importance of the preliminary studies conducted here on plant protein antioxidant properties.

Oxidative stress is considered one of the causes of a variety of acute and chronic diseases, such as cancer, diabetes, cardiovascular disease, and Parkinson’s disease [6]. Therefore, the development of antioxidants has become an important direction of the pharmaceutical field. Plant proteins demonstrate multiple biological activities, such as antioxidant [7], antitumor [8], immunomodulatory [9] and hypoglycemic activity [10]. There are some antioxidant proteins that have been extracted from plant material, such as seeds from Chinese chives [11], *camellia* [12], and *Toona sinensis* [13].

The constituents of *Semen Allii Fistulosi* (PSAF) are carbohydrates, proteins, fat and oil, palmitic acid adenosine, S-(*cis*-1-propenyl)-*L*-cysteine, β-sitosterol, etc [1,4]. The aim of the present work was to investigate the extraction of proteins from PSAF and preliminarily characterize the physicochemical properties and in vitro antioxidant activity of extracted proteins. In order to facilitate the extraction of water-soluble protein, PSAF was degreased during pretreatment. The solubilities of protein are affected by some conditions, such as pH value, ionic strength, temperature, salts, and solvent types. Thus, the proteins of *A. fistulosum* L. seeds were extracted using the alkali water method. This work provides a scientific basis for the study of plant proteins as antioxidants, and has important practical significance and broad market potential.

## 2. Results and Discussion

### 2.1. PSAF Extraction Process Parameters

Proteins are ampholytes with positively and negatively charged groups, and their solubility varies with pH. Thus, the appropriate pH value is helpful to protein extraction. With respect to the effect of pH on proteins from *Semen Allii Fistulosi* (PSAF) extraction rate, as shown in Figure 1A, when pH values were between 5 and 8, the yield was observed to rise to a peak at pH 8.0. When pH exceeded 8.0, yields began to decrease, indicating the system was getting closer to isoelectric point for PSAF, which reduced yields by bringing about a decrease in the solubility. Thus, the ideal pH was determined to be pH = 8.0.

As shown in Figure 1B, the effect of extraction time on PSAF extraction rate was investigated in this study. The yield of PSAF increased with the extraction time, reaching a maximum at 150 min, after which showed slight decreases in yield. The length of time affects extraction efficiency and yield of proteins. The proteins denaturation caused the yield to decrease for a long duration [14]. Thus, the optimal extraction time was chosen as 150 min.

Different ratios of liquid-to-solid affect solution viscosity, molecular diffusion, and protein dissolution [15,16]. With respect to the effect of liquid to solids ratio on extraction, as shown in Figure 1C, the PSAF yield rose clearly with an increasing ratio of liquid to material over the range of 10 to 30 mL/g, reaching a maximum value at a ratio of 40 mL/g. After that, yields stabilized. Considering the experimental cost and extraction efficiency, the optimal liquid material ratio was chosen as 30 mL/g.

The extraction temperature can affect the protein yield [11], which may cause the protein steric structure to stretch, and the molecular thermal motion to intensify. High temperatures may destroy the spatial conformation of the protein and may cause the natural conformation of the protein to disintegrate.

As shown in Figure 1D, when the temperature was raised from 30 °C to 60 °C, the yield was increased until it reached the maximum. However, when the temperature exceeded 60 °C, the yield was lowered. Based on the extraction efficiency and protein stability, 60 °C was advisable to select an extraction temperature.

### 2.2. Optimization for PSAF Extraction

The results of the orthogonal test and extreme difference analysis are shown in Table 1. The influence by the factors on the extraction yield of PSAF increased in the order of: B (extraction time) < A (pH value) < D (temperature) < C (liquid-to-solid ratio) according to the R values. Based on this analysis, and considering the PSAF extraction efficiency, the cost of energy and the feasibility of experiment, the optimum conditions of extraction were determined as follows: extraction time 150 min, extraction pH 8.5, extraction temperature 60 °C, and ratio (*v*/*w*, mL/g) of extraction solvent to raw material 35.

### 2.3. Characterization of PSAF

To more accurately determine the isoelectric point of the PSAF, values from pH 2 to pH 12 were sequentially set. As shown in Figure 2A, two distinct troughs were demonstrated around pH values of 5 and 10, indicating that there were different isoelectric points for PSAF. Subsequently, eight pH values at values intervals of 0.4 were taken near these two pH values for further detection. As shown in Figure 2B,C, the absorbance was minimal at pH 4.4 and pH 10.2, and isoelectric points for PSAF were identified to be 4.4 and 10.2. As the isoelectric point represents pH values wherein a given protein has its lowest solubility, this information is critical when determining protein utilization, especially in food processing [17,18].

With respect to amino acid composition, as shown in Table 2 and Figure 3, PSAF contained 17 different amino acids, and the species of amino acids were found to be relatively comprehensive, while isoleucine and leucine were found to be the limiting amino acids for PSAF. Glutamic acid was the highest among the whole amino acids, and the proportion of sulfur-containing amino acids, aromatic amino acids, and lysine in total amino acids was higher than the FAO/WHO reference value. 

However, glutamic acid could combine with blood ammonia and transform into harmless glutamine, which participates in brain tissue metabolism and improves brain function [19]. The frequency of developmental issues in children increases due to a lack of dietary lysine [20]. Therefore, according to the protein complementation theory [21], PSAF could be used as a food fortifier to complement other proteins and improve the nutritional value of various foods.

SDS-PAGE has become a widely used method for protein separation [22], and as the results show in Figure 4, three distinct bands were separated on gel. A darker band was located between 40 kDa and 55 kDa, indicating that most of the PSAF protein was concentrated in this range, in comparison to the other two weaker bands at values of 15 kDa and 10 kDa.

As shown in Figure 5, FT-IR analysis shows the absorption peak of N-H stretching vibration was in the range of 3400–3440 cm^−1^. The amide A band of SPAF appeared at 3291 cm^−1^, indicating that hydrogen bonds were formed between N-H and C=O. The bands appearing at 2860–2930 cm^−1^ were due to the symmetric and asymmetric stretching vibration of C-H found in the aliphatic chain of proteins and lipids [23]. The absorption bands appearing at 1655, 1539, and 1395 cm^−1^ were assigned to the amides I, II, and III, respectively [24]. Amide I was caused by the stretching vibration of C=O and which was often used for secondary structure analysis of proteins. Amide II mainly derived from N-H bending vibration and followed by C-N stretching vibration.

The bands observed at 1452 and 1242 cm^−1^ were attributed to the absorption of CH_2_ bending (scissors) vibration and C-N stretching modes [25]. The absorption peak at 1308 cm^−1^ vibrates at a range attributed to in-plane hydroxyl deformation [26]. The bands around 1168 and 1080 cm^−1^ may be caused by antisymmetric stretching variation of C-O-C [25].

### 2.4. Antioxidant Activities in vitro

DPPH represents a very stable free radical centered on nitrogen, and is widely used in the study of free radical scavenging ability of various antioxidants [27]. Figure 6A shows that DPPH scavenging ability increased within lower scope of concentration (0–2 mg/mL). The IC_50_ value is 1.43 mg/mL. The change in antioxidant capacity was not obvious and tended to be stable at a concentration higher than 2 mg/mL, which may be due to the ethanol solution restricted protein dissolution. The solubility of VC was relatively high, which may be one of the reasons why the scavenging effect of VC was better than that of PSAF.

Hydroxyl radicals can interact with different molecules in cells and cause oxidative damage to sugars, amino acids, nucleic acids, and lipids [28]. In terms of hydroxyl scavenging, Figure 6B shows that there was a dose-dependent relationship between scavenging rate and concentration, and the scavenging rate rose with the increasing of concentration, with an IC_50_ value for PSAF of 1.37 mg/mL. Interestingly, at certain concentrations, the scavenging rate of PSAF was higher than that of VC.

Superoxide radicals can be metabolized to hydroxyl radicals by peroxidase in the body to hydroxyl free radicals, causing chronic and senile diseases [29]. The superoxide radical scavenging activity of PSAF was correlated with the concentration, and there was a tendency to approach or even exceed VC. At the maximum concentration of 5 mg/mL, the clearance rate reached 88.76 ± 0.55%, and the IC_50_ value was 2.17 mg/mL, indicating that PSAF has a certain ability to scavenge superoxide (Figure 6C).

Fe^2+^ can trigger free radical formation and thus cause lipid peroxidation or DNA damage [30]. The compounds with strong chelation ability for Fe^2+^ usually had functional groups, such as −OH, −SH, −COOH, C=O, −NR_2_, and −S [31,32]. The protein was provided with similar groups, which had the potential metal chelating capacity. With respect to Fe^2+^ chelating activity, as shown in Figure 6D, within low concentrations (<1 mg/mL), PSAF showed a strong ability to chelate Fe^2+^, which was superior to that of VC. In particular, at a concentration of 0.5 mg/mL, the chelation rate had exceeded 99% and the IC_50_ value was 0.006 mg/mL.

As shown in Figure 6E, the reducing power increased with the increasing of concentration. At the maximum concentration of 10 mg/mL, the OD_700_ value was close to 0.5. However, there was still a noticeable difference in reducing power between the PSAF and VC. Determination of reducing power is an important part of the antioxidant activity test. The greater the absorbance, the greater the reducing power and the antioxidant capacity.

## 3. Materials and Methods

### 3.1. Materials and Equipment

High-performance liquid chromatograph (Agilent 1260, Agilent Technology Ltd., Palo Alto, CA, USA), an ultraviolet spectrophotometer (UV-800A) from Shanghai Metash instruments Ltd. (Shanghai, China), and a Fourier transform infrared spectrometer (Nicolet is10) from Thermo Fisher Scientific Ltd. (Waltham, MA, USA) were used. Samples were freeze-dried with an FDU-1100 model from Eyela Ltd. (Tokyo, Japan). For molecular experiments, a double-sided vertical electrophoresis tank (JY-SCZ2, Beijing Junyi Electrophoresis Ltd., Beijing, China) and electrophoresis apparatus (DYY-6C, Beijing LiuYi Instrument Factory, Beijing, China) were used.

*Semen Allii Fistulosi* was obtained from Weifang Jian Xin pharmacy chain Co., Ltd. (Weifang, China). Coomassie brilliant blue G250 (Amresco 0615) and bovine serum albumin were obtained from Wako Pure Chemical Industries Ltd. (Osaka, Japan). The protein molecular weight markers were obtained from ThermoFisher Scientific Ltd. (Waltham, MA, USA). The standard reagents for analyzing amino acids were obtained from Sigma Ltd. (St. Louis, MO, USA). All other reagents used in the experiments were of analytical grade.

### 3.2. Sample Pretreatment

Dried raw material was ground into a powder and then pretreated with petroleum ether to degrease. The solid residue was separated from the solvent with filtration and dried at room temperature.

### 3.3. Protein Content Determination

According to the Coomassie Brilliant Blue method [33], a series of known concentrations (0.01, 0.03, 0.05, 0.07 and 0.09 mg/mL) of bovine serum albumin were used to establish the standard curve. The obtained equation was y = 5.2794x + 0.0052, coefficient of determination r^2^ = 0.9996, where x was the concentration and y was the absorbance. The PSAF yield was calculated with the formula, yield (%, *w*/*w*) = (*m_1_*/*m_2_*) × 100, where m_1_ was the mass of PSAF, and m_2_ was the mass of dried sample.

### 3.4. Extraction Optimization

To determine the effects of pH, extraction time, liquid-to-solid ratio, and temperature on extraction, single-factor extraction experiments were performed under the following conditions: pH of 5.0, 6.0, 7.0, 8.0, 9.0, and 10.0, extraction time of 30, 60, 90, 120, 150, and 180 min, liquid-to-solid ratio of 10:1, 20:1, 30:1, 40:1, and 50:1, temperature of 30, 40, 50, 60, and 70 °C. The effect of each factor on the yield was assessed to determine the optimum level. The initial extraction parameters were as follows: pH 8, 120 min extraction time, liquid-to-solid ratio of 20 mL/g, and 40 °C extraction temperature.

The raw material (5 g) was extracted with 0.02 M phosphate buffer under the conditions described above. After extraction, the extracted solution was filtered and centrifuged. Then, the supernatant was concentrated at 60 °C under vacuum. Ammonium sulfate was added to the concentrated liquid to a final concentration of 95% and kept overnight at 4 °C. After centrifugation, dialysis, and lyophilization, the crude PSAF was obtained.

The pH value, extraction time, liquid-to-solid ratio, and extraction temperature were selected based on the results of the single factor exploration experiment. An L_9_(_3_^4^) orthogonal table was used to confirm the extraction process, and the factors and levels are shown in Table 3.

### 3.5. Characterization of PSAF Isoelectric Point and Amino Acid Composition

To determine the isoelectric point, the PSAF solution was adjusted to different pH levels and centrifuged [34]. The protein content in the solution was measured, and the pH at the trough was determined. First, eleven integer pH values (pH 2–12) were selected to determine. Then, 8 pH values with a spacing of 0.4 were selected near the trough pH for measurement. Finally, the pH at the bottom of the curve was the isoelectric point of the protein.

Amino acids composition was determined according to the method [33]. PSAF was hydrolyzed with 6 M HCl at 110 °C for 24 h in a sealed tube with phenyl isothiocyanate (PITC) precolumn derivatization. The amino acids composition was analyzed using high-performance liquid chromatography (Agilent, USA). The chromatographic conditions were as follows: mobile phase (gradient elution), A: acetonitrile-0.1 M sodium acetate solution (3:97, *v*/*v*); B: acetonitrile-water (4:1, *v*/*v*); column, Shiseido C18 (4.6 × 250 mm, 5 μm); flow rate, 1.0 mL/min; column temperature, 40 °C; detector wavelength, 245 nm.

### 3.6. SDS-PAGE and FT-IR Spectra Analysis

SDS-PAGE was run according to a previously described method [35]. PSAF samples were dissolved in distilled water, and the solutions were mixed with 20 μL of sample buffer and heated at 100℃ for 10 min. Sample and marker were loaded onto precast polyacrylamide gel (resolving gel of 15% and stacking gel of 5%). The electrophoresis experiments were run at 120 V for 35 min and subsequently ran at 150 V until the dye reached the bottom of the gel. The gel was stained with 0.1% Coomassie-Blue G-250.

Infrared (IR) analysis was performed using a Fourier transform infrared spectrophotometer. Approximately 2 mg of the dried sample was mixed with an appropriate amount of KBr, ground uniformly, and tableted using a tableting machine. IR spectrum was recorded in the frequency range of 4000–500 cm^−1^ [36].

### 3.7. DPPH Radical Scavenging Assay

PSAF was dissolved in deionized water to prepare a range of concentrations (0.05, 0.1, 0.5, 1, 2, 3, 4 and 5 mg/mL) and 200 μL sample aliquots were taken and measured with 200 μL of DPPH (0.4 mM in dehydrated alcohol). The absorbance was measured at 517 nm, and Vitamin C (VC) was used as positive control [37]. DPPH radical scavenging activity was calculated using the following equation: Scavenging rate (%) = [1 − (A_2_ − A_1_)/A_0_] × 100, where A_0_ was the absorbance measured when the sample solution was replaced with distilled water, A_1_ was the absorbance measured when DPPH was replaced with dehydrated alcohol, and A_2_ represents absorbance measured in the sample solution.

### 3.8. Hydroxyl and Superoxide Radical Scavenging Activity

The hydroxyl radical content of the reaction solution can be detected at 520 nm [38]. VC was used as the positive control. Briefly, the reaction mixture contained 200 μL sample solution, 200 μL phosphate buffer (1.5 mM, pH 7.4), 200 μL safranine T solution (360 μL/mL), 100 μL EDTANa_2_-Fe solution (12 mM) and 200 μL H_2_O_2_ solution (3%, *v*/*v*). The reaction solution was placed at 37 °C for 30 min. Hydroxyl radical scavenging activity was calculated by equation: Scavenging rate (%) = (A_2_ − A_1_)/(A_0_ − A_1_) × 100, where A_0_ was the absorbance measured when the sample solution and H_2_O_2_ solution was replaced with distilled water, A_1_ was the absorbance measured when the sample solution was replaced with distilled water, and A_2_ was the absorbance measured of the sample solution.

The scavenging ability of superoxide radical was measured by the slightly modified method [39]. Tris-HCl buffer (16 mM, pH 8.0) was used as a reaction solvent. 300 μL of the sample solution was mixed with NADH solution (338 μM), NBT solution (72 μM), and PMS (30 μM) of 50 μL respectively for 5 min. The absorbance was measured at 560 nm. VC was used as the positive control. The superoxide radical scavenging activity was calculated using the following equation: Scavenging rate (%) = [1 − (A_2_ − A_1_) / A_0_] × 100, where A_0_ was the absorbance measured when the sample solution was replaced with Tris-HCl buffer, A_1_ was the absorbance measured when the NBT was replaced with Tris-HCl buffer, A_2_ was the measured absorbance of the sample solution.

### 3.9. Fe^2+^ Capacity Activity and Reducing Power Determination

The extent of Fe^2+^ chelation was measured by the described method [40]. In brief, 200 μL sample aliquots were mixed with 20 μL FeCl_2_ solution (2 mM) and 40 μL ferrozine solution (5 mM) for 10 min, and the absorbance at 562 nm was measured, with EDTA-2Na used as the positive control. The chelating activity (%) was calculated using the following equation: Chelating rate (%) = [1 − (A_2_ − A_1_)/A_0_] × 100, where A_0_ was the absorbance measured when the sample solution was replaced with distilled water, A_1_ was the absorbance measured when the FeCl_2_ was replaced with distilled water, and A_2_ was the absorbance of the sample solution.

The determination of reduction ability was based on the method by reference [41]. Two hundred microliters of phosphate buffer, 60 μL sample aliquots, and 200 μL potassium ferricyanide solution (1%, *w*/*v*), were combined, and reacted at 50 °C for 20 min. Two hundred microliters of trichloroacetic acid solution (10%, *w*/*v*) was added to terminate the reaction. Two hundred microliters of reaction solution were taken, and 200 μL of distilled water and 40 μL FeCl_3_ solution (0.1%, *w*/*v*) were added, mixed, and after 10 min of reaction at room temperature the absorbance was measured at 700 nm. VC served as a positive control.

### 3.10. Statistical Analysis

SPSS 17.0 software (SPSS Inc., Chicago, IL, US) was used for orthogonal design and its variance analysis used to determine a significant difference at *p* < 0.05. The data are presented as mean ± standard deviation (SD). All experiments were repeated three times.

## 4. Conclusions

In this study, the orthogonal design was used to optimize the extraction of protein from *Semen Allii Fistulosi* (PSAF) and after that, the protein extract was prepared by determined extraction conditions and ammonium sulfate precipitation. Then, isoelectric point measurement, HPLC, SDS-PAGE, and FT-IR were studied for preliminary characterization of PSAF. Finally, the antioxidant activity in vitro of PSAF were determined. These findings suggest that PSAF has potential antioxidant effects, and more study is merited.

## Figures and Tables

**Figure 1 molecules-23-03235-f001:**
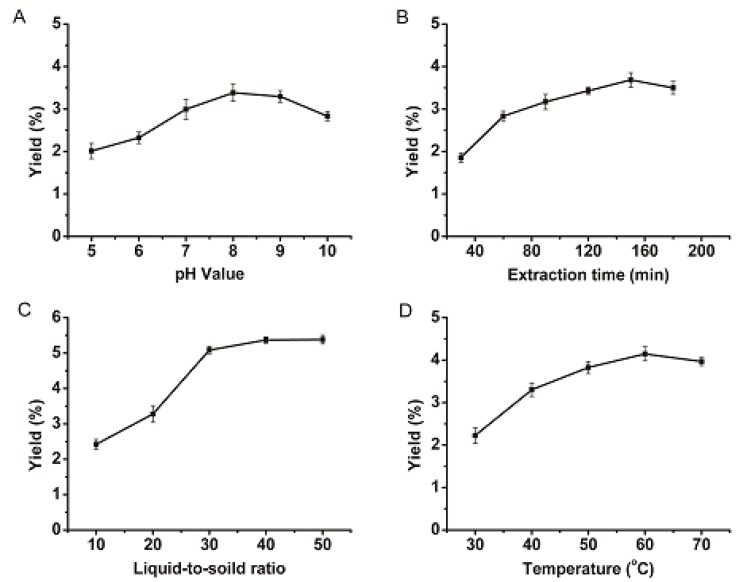
The effect of different extraction parameters: pH value (**A**), extraction time (**B**), liquid-to-solid ratio (**C**), temperature (**D**) on yield of PSAF.

**Figure 2 molecules-23-03235-f002:**
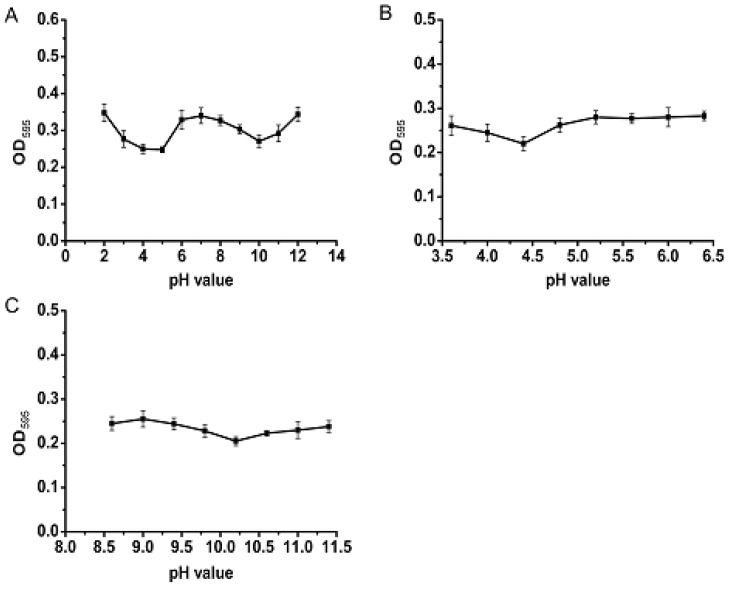
Absorbance of supernatants with different pH after reaction with Coomassie Brilliant Blue. (**A**) pH 2–12; (**B**) pH 3.6–6.4; (**C**) pH 8.6–11.4.

**Figure 3 molecules-23-03235-f003:**
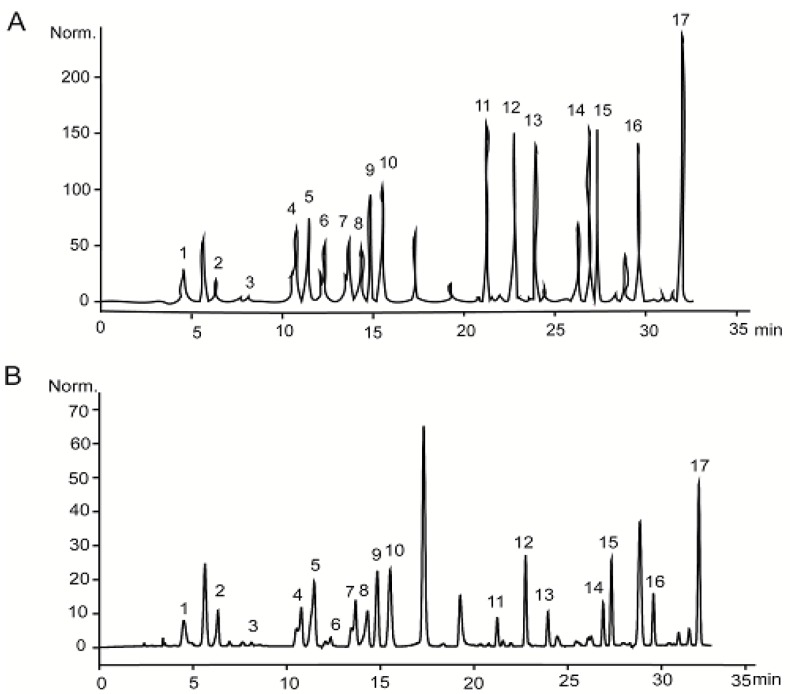
HPLC analysis of amino acids standard (**A**), and PSAF (**B**). Peaks: 1, Aspartic acid; 2, Glutamic acid; 3, Cysteine; 4, Serine; 5, Glycine; 6, Histidine; 7, Arginine; 8, Threonine; 9, Alanine; 10, Proline; 11, Tyrosine; 12, Valine; 13, Methionine; 14, Isoleucine; 15, Leucine; 16, Phenylalanine; 17, Lysine.

**Figure 4 molecules-23-03235-f004:**
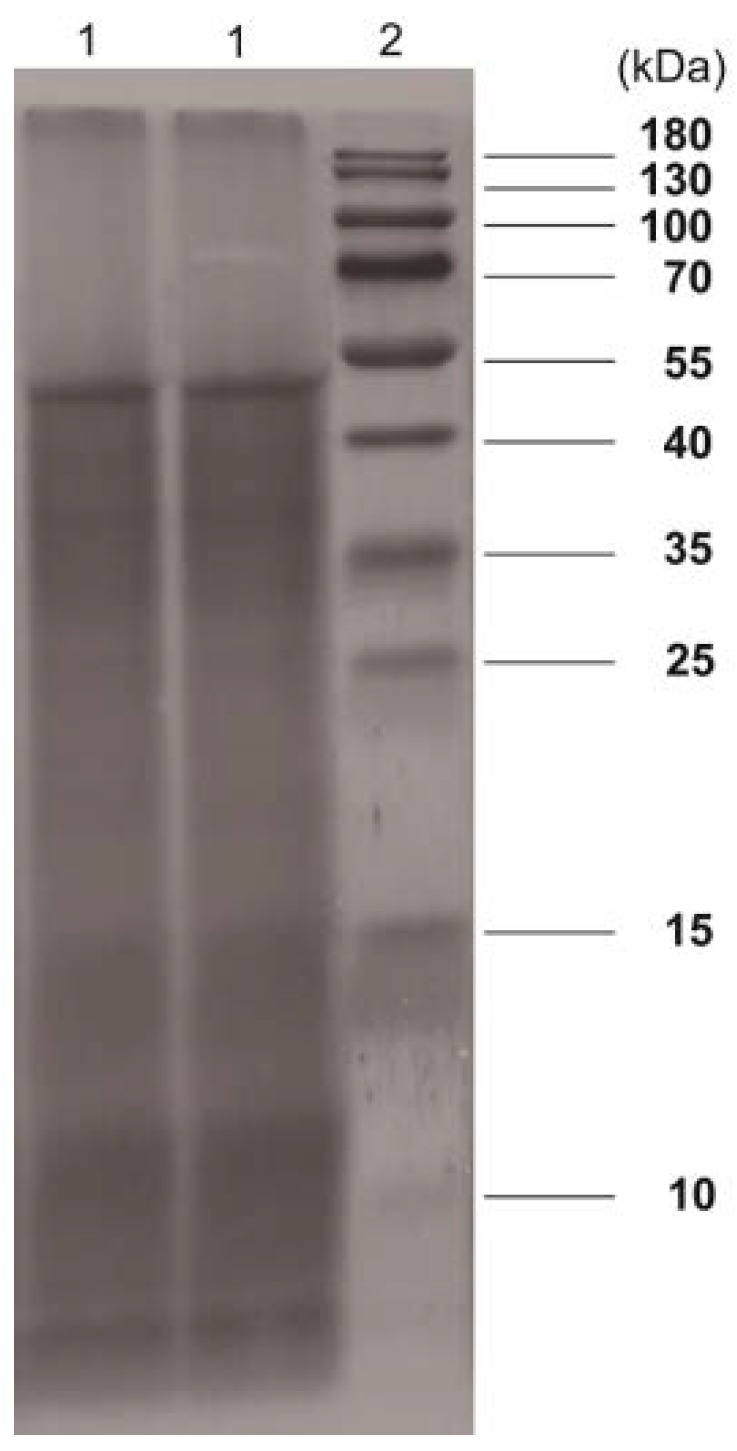
SDS-PAGE analysis of PSAF. Land 1: PSAF; Land 2: standard protein marker.

**Figure 5 molecules-23-03235-f005:**
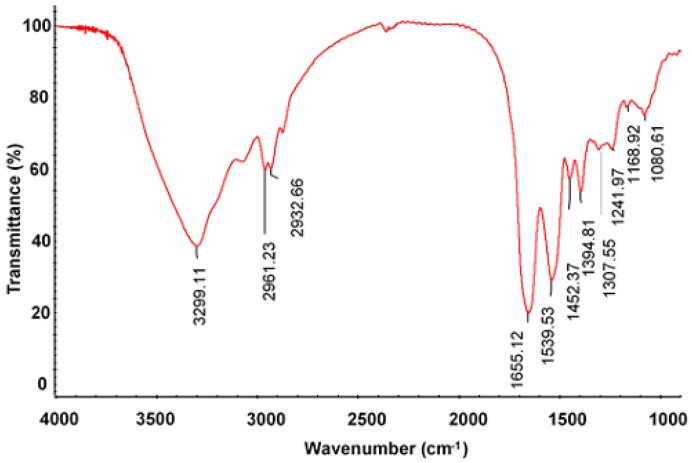
FT-IR spectra of PSAF.

**Figure 6 molecules-23-03235-f006:**
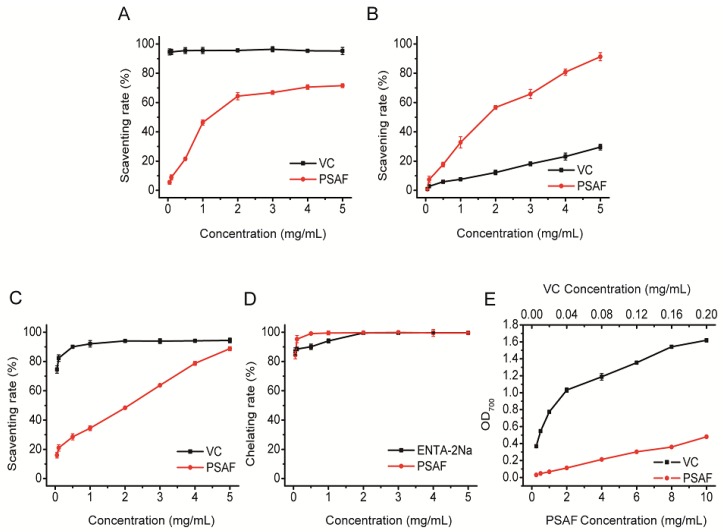
Antioxidant activities of PSAF in vitro. Scavenging effects on DPPH radical (**A**), hydroxyl radical (**B**), superoxide radical (**C**) and Fe^2+^ chelating activity (**D**) and reducing power (**E**).

**Table 1 molecules-23-03235-t001:** Analysis of L_9_(3^4^) test results.

No.	pH (A)	Extraction Time (B)	Liquid-to-Solid Ratio (C)	Temperature (D)	Yield of PSAF (%)
1	1	1	1	1	4.41
2	1	2	2	2	4.86
3	1	3	3	3	6.29
4	2	1	3	3	4.67
5	2	2	1	1	4.94
6	2	3	2	2	4.63
7	3	1	2	2	6.11
8	3	2	3	3	4.75
9	3	3	1	1	4.52
K1	5.19	5.06	4.60	4.62	
K2	4.74	4.85	4.68	5.20	
K3	5.13	5.15	5.78	5.24	
R	0.44	0.30	1.18	0.62	

Ki: the mean of the corresponding levels of protein yield at each factor (i = 1, 2, 3); R: the extreme difference between different levels of a factor.

**Table 2 molecules-23-03235-t002:** Analysis of amino acids composition (mg/100 mg) of PSAF.

Amino Acid	PSAF	FAO/WHO	Amino Acid	PSAF	FAO/WHO
Aspartic acid	6.33	-	Valine	4.49	4.96
Serine	4.47	-	Cysteine & methionine	5.44	3.52
Glutamic acid	15.27	-	Isoleucine	2.01	4.0
Glycine	4.2	-	Leucine	4.43	7.04
Alanine	4.88	-	Phenylalanine & tyrosine	6.74	6.08
Histidine	1.69	-	Lysine	6.22	5.44
Arginine	8.78	-	FLAA	Ile	-
Proline	2.26	-	SLAA	Leu	-
Threonine	2.81	4.0			

FAO: Food and Agriculture Organization; WHO: World Health Organization; FLAA: first limiting amino acid; SLAA: second limiting amino acid.

**Table 3 molecules-23-03235-t003:** Factors and levels for orthogonal design.

Factor Level	pH	Extraction Time (min)	Liquid-to-Solid Ratio	Temperature (°C)
1	8.0	90	25:1	40
2	8.5	120	30:1	50
3	9.0	150	35:1	60

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
