# Peer review of "Extraction, Characterization and Antioxidant Activity in vitro of Proteins from Semen Allii Fistulosi"

_molecules, 2018, doi:10.3390/molecules23123235_

Round 1
Reviewer 1 Report
The manuscript entitled “Extraction, characterization and antioxidant activity in vitro of proteins from Semen Allii Fistulosi” have demonstrated significant results from extraction and characterization of proteins from Allii Fistulosi. The study present significant results in biological properties of phytochemicals fields. The work was written carefully and well in terms of language. The Authors of a well-planned, performed and interpreted research. Moreover, statistical analyzes were also well planned.
Authors should correct manuscript according to the suggestion and completed some information.
Minor issues:
Materials and methods:
In my opinion, the Authors should calculate the IC50 values for antioxidant properties e.g DPPH, hydroxyl and superoxide radicals scavenging and chelating Fe2+. It is a universal unit that allows Authors to compare results with other research.
Page 9, line 246: Authors should give information about percent the polyacrylamide gel?
Page 9, line 262: Authors should complete method modification and complete sample volume and others chemicals as DPPH assay
Page 9, line 278: Authors should complete FeCl2 and ferrozine volume and concentrations
Author Response
Dear Editor and Reviewer:
Thank you for your letter and for the reviewers’ comments concerning our manuscript entitled “Extraction, characterization and antioxidant activity in vitro of proteins from Semen Allii Fistulosi” (ID: molecules-401235).
Those comments are valuable and very helpful for revising and improving our paper, as well as the guiding significance to our researches. We have studied comments carefully and have made correction which we hope meet with approval. Revised portion are marked in colour in the paper.
Specific comments and Suggestions
1. Materials and methods:
Response: “.Materials and Methods” in line 188 was corrected as “Materials and methods”.
2. In my opinion, the Authors should calculate the IC50 values for antioxidant properties e.g DPPH, hydroxyl and superoxide radicals scavenging and chelating Fe2+. It is a universal unit that allows Authors to compare results with other research.
Response: The IC50 values for DPPH, hydroxyl and superoxide radicals scavenging activity are 1.43 mg/mL, 1.37 mg/mL, 2.17 mg/mL, respectively. And the calculated IC50 value of Fe2+ chelating activity is about 0.006 mg/mL, which has been reflected in the manuscript. However, the minimum value of Fe2+ sequestration rate has reached about 84% (>50%) in the range of measured concentration (0.05-5mg/mL). Therefore, the IC50 value obtained by extrapolation have a certain error.
3. Page 9, line 246: Authors should give information about percent the polyacrylamide gel?
Response: “Sample and marker were loaded onto precast polyacrylamide gel” was corrected as
“Sample and marker were loaded onto precast polyacrylamide gel (resolving gel of 15% and stacking gel of 5%)”.
4. Page 9, line 262: Authors should complete method modification and complete sample volume and others chemicals as DPPH assay.
Response: The line 262-267 was corrected as "The hydroxyl radical content of the reaction solution can be detected at 520 nm [38].VC was used as the positive control. Briefly, the reaction mixture contained 200 μL sample solution, 200 μL phosphate buffer (1.5 mM, pH 7.4), 200 μL safranine T solution (360 μL/mL), 100 μL EDTANa2-Fe solution (12 mM) and 200 μL H2O2 solution (3%, v/v). The reaction solution was placed at 37℃ for 30 min....”.
5. Page 9, line 278: Authors should complete FeCl2 and ferrozine volume and concentrations
Response: “...200-μL sample aliquots were mixed with FeCl2 and ferrozine for 10 min...” was corrected as “...200 μL sample aliquots were mixed with 20 μL FeCl2 solution (2 mM) and 40 μL ferrozine solution (5 mM) for 10 min...”.
We tried our best to make some modifications in the manuscript. These changes will not influence the content and framework of the paper. We appreciate for Editor/Reviewers’ warm work, and hope that the correction will meet with approval. Once again, thank you very much for your comments and suggestions.
Sincerely yours,
Wanzhong Li
December 1, 2018
Reviewer 2 Report
Please find all my questions, suggestions, corrections, etc., in enclosed text of manuscript.

Author Response
Dear Editor and Reviewer:
Thank you for your letter and for the reviewers’ comments concerning our manuscript entitled “Extraction, characterization and antioxidant activity in vitro of proteins from Semen Allii Fistulosi” (ID: molecules-401235).
Those comments are valuable and very helpful for revising and improving our paper, as well as the guiding significance to our researches. We have studied comments carefully and have made correction which we hope meet with approval. Revised portion are marked in colour in the paper.
Responds to the reviewer’s comments:
Reviewer #2:
1.Response to comment: What is OD? Maybe A?
Response: The optical density is a logarithm of the ratio of the incident light intensity to the transmitted light intensity. And the absorbance is the logarithm of the ratio of the transmittance of incident and transmitted light, also known as optical density, which shows that the two interpretations are in fact consistent. That is to say, “OD” is equivalent to “A”.
2.Response to comment: What is -?
Response: “-” means that FAO/WHO has no clear regulations on the content of certain amino acids.
3.Response to comment: marjer is marker?
Response: The “marjer” is corrected as "marker".
4.Response to comment: FI-IR is FT-IR?
Response: The “FI-IR” is corrected as “FT-IR”.
5.Response to comment: SFP is PSAF or something else?
Response: The “SFP” is corrected as “PSAF”.
6.Response to comment: What is IC50 value?
Response: IC50 is the drug concentration causing 50% inhibition of the desired activity. The IC50 value in this paper refer to the concentration of protein sample when the free radical scavenging rate reaches 50%.
7.Response to comment: What about mean particle size of powder?
Response: We smashed dried raw material to a particle size that can pass through the 40 mesh sieve.
8.Response to comment: What about boiling point?
Response: Since the proteolysis reaction is carried out in a sealed tube, even if the boiling point of hydrochloric acid is reached, the reaction solution does not splash outside. This method was also used in the following literatures:
[1] Martín-Hernández, C.; Bénet, S. Determination of proteins in refined and nonrefined oils. J. Agric. Food. Chem. 2008, 56, 4348-4351, doi:10.1021/jf7036888.
[2] Tan, E.S; Ying-Yuan, N. A comparative study of physicochemical characteristics and functionalities of pinto bean protein isolate (PBPI) against the soybean protein isolate (SPI) after the extraction optimisation. Food. Chem. 2014, 152, 447-455, doi:10.1016/j.foodchem.2013.12.008.
[3] Gaurav Kumar, P.; Nidheesh, T.; Govindaraju, K.; Jyoti; Suresh, P.V. Enzymatic extraction and characterisation of a thermostable collagen from swim bladder of rohu (Labeo rohita). J. Sci. Food. Agric. 2017, 97, 1451-1458, doi:10.1002/jsfa.7884.
9.Response to comment: Gradient conditions?
Response: The elution gradient is as follows:
10.Response to comment: Is this reference correct?
Response: Yes, it is correct.
11.Response to comment: Solution? Concentration?
Response: Yes, it's a solution. “...and 200 μL potassium ferricyanide, were combined, and reacted at 50℃ for 20 min. Two hundred microliters of trichloroacetic acid was added to terminate the reaction. Two hundred microliters of reaction solution was taken, and 200 μL of distilled water and 40 μL FeCl3 were...” was corrected as “...and 200 μL potassium ferricyanide solution (1%, w/v), were combined, and reacted at 50℃ for 20 min. Two hundred microliters of trichloroacetic acid solution (10%, w/v) was added to terminate the reaction. Two hundred microliters of reaction solution was taken, and 200 μL of distilled water and 40 μL FeCl3 solution (0.1%, w/v) were...”
Improved place
1.Response to comment: ...known concentrations (from ? to ?) with...
Response: “...known concentrations with...” was corrected as “... known concentrations (0.01, 0.03, 0.05, 0.07 and 0.09 mg/mL) with...”
2.Response to comment: ...a range of concentrations (from ? to ?) and 200 μL sample...
Response: “...a range of concentrations and 200 μL sample...” was corrected as “...a range of concentrations (0.05, 0.1, 0.5, 1, 2, 3, 4 and 5 mg/mL) and 200 μL sample...”
3.Response to comment: H2O2?
Response: “H2O2” was corrected as “H2O2 solution”
4.Response to comment: ...μL FeCl3 solution (concentration?) were...
Response: “...40 μL FeCl3...” was corrected as “...40 μL FeCl3 solution (0.1%, w/v) ...”
Wrong writing and formatting
1.“Semen Allii Fistulosi” is a Latin name. Sorry this is our fault. “Semen Allii Fistulosi” in lines 3, 25 was corrected as “Semen Allii Fistulosi”
“Semen Allii Fistulosi” in lines 10, 12, 37, 47, 51, 197, 296 was corrected as “Semen Allii Fistulosi”
2.“in vitro” in lines 3, 11, 23, 50, 298 was corrected as “in vitro”
3.“...at different pH” in line 16 was corrected as “...at different pH values. ”
4.“...while chelating Fe2+ activity, and demonstrating weak reducing power.” in line 22 was corrected as “... while chelating Fe2+ activity and demonstrating weak reducing power.”
5.“...S-(cis-1-propenyl)...” in line 48 was corrected as “...S-(cis-1-propenyl)... ”
6.“...β-sitosterol etc [1,4].” in line 48 was corrected as “...β-sitosterol, etc [1,4]. ”
7.“...investigate the extraction of PSAF and...” in line 49 was corrected as “...investigate the extraction of proteins from Semen Allii Fistulosi (PSAF) and...”
8.“...protein, Semen Allii Fistulosi (PSAF) was...” in line 51 was corrected as “...protein, Semen Allii Fistulosi was ...”
9.“Thus, the protein of A. fistulosum L. seed was...” in line 53 was corrected as “Thus, the proteins of A. fistulosum L. seeds were...”
10.“...of pH on PSAF extraction rate, ...” in line 60 was corrected as “...of pH on proteins from Semen Allii Fistulosi (PSAF) extraction rate, ...”
11.“...yield of protein. The protein denaturation...” in line 67 was corrected as “...yield of proteins. The proteins denaturation...”
12.“...extraction parameters pH (A), ...” in line 86 was corrected as “...extraction parameters: pH value (A), ...”
13.“...temperature(D) on yields of PSAF.” in line 87 was corrected as “...temperature (D) on yield of PSAF.”
14.“the extraction yields of PSAF...” in line 90 was corrected as “...the extraction yield of PSAF...”
15.“Analysis of L9(34) test results.” in line 96 was corrected as “Analysis of L9(34) test results.”
16.“...in Figure 2B, 2C, the...” in line 104 was corrected as “...in Figures 2B and 2C, the...
17.Figure 2C was corrected as required.
18.“However, Glutamic acid...” in line 117 was corrected as “However, glutamic acid...”
19.“Analysis of amino acid composition...” in line 122 was corrected as “Analysis of amino acids composition...”
20.“Figure 3. Chromatographic analysis of amino acid standard...” in line 127 was corrected as “Figure 3. HPLC analysis of amino acids standard...”
21.“Vc” in lines 158, 159, 165, 169, 177, 181, 256, 262, 270, 289 was corrected as “VC”.
22.Figure 6 was corrected. “Vc” was corrected as “VC”; “SFP” was corrected as “PSAF”.
23.“3.Materials and Methods” in line 188 was corrected as “3. Materials and methods”.
24.“High-performance liquid chromatography (Agilent 1260, Agilent Technology Ltd, USA), while an ultraviolet spectrophotometer (UV-800A) was obtained from Shanghai” in line 190-191 was corrected as “High-performance liquid chromatograph (Agilent 1260, Agilent Technology Ltd, USA), an ultraviolet spectrophotometer (UV-800A) from Shanghai...”.
25.“... (USA) was used. ” in line 193 was corrected as “... (USA) were used.”
26.“... (Amresco 0615), bovine serum...” in line 198 was corrected as “... (Amresco 0615) and bovine serum... ”
27.“The regression line was y = 5.2794x + 0.0052, r2 = 0.9996, where x was...” in line 209-210 was corrected as “The obtained equation was y = 5.2794 x + 0.0052, coefficient of determination r2 = 0.9996, where x was...”
28.“...yield (%) = (w/W) × 100%, where w was the mass of PSAF, and W was the mass of dried sample.” in line 211-212 was corrected as “...yield (%, w/w) = (m1/m2) × 100, where m1 was the mass of PSAF, and m2 was the mass of dried sample.”
29.“...exploration experiments. An L9(34) orthogonal...” in line 227 was corrected as “...exploration experiment. An L9(34) orthogonal...”
30.“Factor Level” in line 299-230 was corrected as “Factor level”.
31.“Amino acid composition...” in line 236 was corrected as “Amino acids composition...”
32.“The amino acid composition...” in line 238 was corrected as “The amino acids composition...”
33.“...follows: mobile phase (gradient elution), A: with acetonitrile-0.1 M sodium acetate solution (3:97); B: acetonitrile-water (4:1); column, Shiseido C18 (4.6 × 250 mm × 5 μm); flow rate, 1.0 mL/min;... ” in line 239-241 was corrected as “...follows: mobile phase (gradient elution), A: acetonitrile - 0.1 M sodium acetate solution (3:97, v/v); B: acetonitrile-water (4:1, v/v); column, Shiseido C18 (4.6 × 250 mm, 5 μm); flow rate, 1.0 mL/min; ...”
34.“...the following formula: scavenging rate (%) = [1 − (A2 − A1) / A0] × 100%, ...” in line 257 was corrected as “...the following equation: Scavenging rate (%) = [1 − (A2 − A1) / A0] × 100, ... ”
35.“..by the formula: scavenging rate (%) = (A2 − A1)/(A0 − A1) × 100%, where ...” in line 263-264 was corrected as “...by equation: Scavenging rate (%) = (A2 − A1) / (A0 − A1) × 100, where...”
36.“... the following formula: scavenging rate (%) = [1 − (A2 − A1) / A0] × 100%, where ...” in line 272 was corrected as “...the following equation: Scavenging rate (%) = [1 − (A2 − A1) / A0] × 100, where...”
37.“Fe2+ capacity” in line 276 was corrected as “Fe2+ capacity”
38.“200-μL”in line 277 was corrected as “200 μL”
39.“...the following formula: chelating rate (%) = [1 − (A2 − A1) / A0] × 100%, where...” in line 280 was corrected as “...the following equation: Chelating rate (%) = [1 − (A2 − A1) / A0] × 100, where...”
40.“...on the method... [41]” in line 284 was corrected as “...on the method by reference [41].”
41.“...60-μL...” in line 285 was corrected as “...60 μL...”
42.“... from Semen Allii Fistulosi, and the protein was prepared by extraction conditions and...” in line 296 was corrected as “...from Semen Allii Fistulosi (PSAF) and after that, the protein extract was prepared by determined extraction conditions and...”
About references
1.You have to correct italic letters and numbers (i.e. missing numbers of journal Vol., pages, etc.).
Response: Italic and missing numbers have been corrected.
We tried our best to make some modifications in the manuscript. These changes will not influence the content and framework of the paper. We appreciate for Editor/Reviewer’s warm work, and hope that the correction will meet with approval. Once again, thank you very much for your comments and suggestions.
Sincerely yours,
Wanzhong Li
December 1, 2018
